# The Effect of Temperature-Assisted High Hydrostatic Pressure on the Survival of *Alicyclobacillus acidoterrestris* Inoculated in Orange Juice throughout Storage at Different Isothermal Conditions

Patra Sourri [1,2], Anthoula A. Argyri [1], George-John E. Nychas [2], Chrysoula C. Tassou [1] and Efstathios Z. Panagou [2,*]

[1] Institute of Technology of Agricultural Products, Hellenic Agricultural Organization DIMITRA, Sof. Venizelou 1, 14123 Lycovrissi, Greece; patrapsourri@gmail.com (P.S.); anthi.argyri@gmail.com (A.A.A.); ctassou@nagref.gr (C.C.T.)

[2] Laboratory of Microbiology and Biotechnology of Foods, Department of Food Science and Human Nutrition, School of Food and Nutritional Sciences, Agricultural University of Athens, Iera Odos 75, 11855 Athens, Greece; gjn@aua.gr

\* Correspondence: stathispanagou@aua.gr

**Abstract:** The purpose of this work was to investigate the population dynamics of the spores and vegetative cells of *A. acidoterrestris* in orange juice treated with temperature-assisted HHP and stored in different isothermal conditions. For this reason, the spores of two *A. acidoterrestris* strains were inoculated in commercial orange juice and subjected to HHP treatment at 600 MPa/60 °C for 5 and 10 min. Inoculated samples were subsequently stored at 4, 12 and 25 °C for 60 days. During storage, the population of *A. acidoterrestris* was determined before and after heat shock at 80 °C for 10 min in order to estimate the quantity of spores and any remaining vegetative cells on the *Bacillus acidoterrestris* medium agar. Results showed that spore populations decreased by 3.0–3.5 log cycles directly after HHP treatment. Subsequently, no significant changes were observed throughout storage regardless of temperature and bacterial strain. However, at 25 °C, an increase of 0.5–1.0 log cycles was noticed. For the remaining vegetative cells, the results illustrated that HHP treatment could eliminate them during storage at 4 and 12 °C, whereas at 25 °C inactivation was strain-dependent. Therefore, temperature-assisted HHP treatment could effectively inactivate *A. acidoterrestris* spores in orange juice and ensure that the inhibitory effect could be maintained throughout storage at low temperatures.

**Keywords:** *Alicyclobacillus acidoterrestris*; high hydrostatic pressure processing; orange juice; shelf life





## 1. Introduction

Nowadays, consumers demand healthier, minimally processed and fresh-like products; therefore, fruit juices which comply with the above requirements have an important place in the human diet [1,2], with orange juice being undoubtedly the most popular fruit beverage worldwide. Although fruit juices are considered to have a prolonged shelf life, the survival of several pathogenic and spoilage microorganisms could limit their marketability and pose serious health issues to consumers [3–5].

The species of *Alicyclobacillus* spp. are the most dominant spore-forming bacteria in the fruit juice industry. *Alicyclobacillus acidoterrestris* in particular is the most important, since it has been considered as a reference microorganism in fruit juice pasteurization due to its high isolation incidence from spoiled products [4,6–9]. *A. acidoterrestris* is a Gram-positive, thermo-acidophilic, non-pathogenic and endospore-forming bacterium. It can grow in a temperature range of 25–60 °C and in a wide pH range from 2.0 to 7.0 [10,11]. The spoilage by *A. acidoterrestris* is characterized by off-flavors described as phenolic, medicinal and

antiseptic due to the production of the chemical spoilage compounds 2-methoxyphenol (guaiacol), 2,6 dibromophenol and 2,6 dihlorophenol [12–15]. The predominant compound indicative of juice spoilage is guaiacol, which is produced by the non-oxidative decarboxylation of vanillic acid and other natural fruit juice components [16]. Spoilage detection is difficult since there is no appearance of acid or gas production leading to swelling of the containers [17].

The fruit juice industry usually controls microbiological growth with pasteurization processes (88–90 °C for 2 min or 90–95 °C for 30–60 s). However, the inactivation of *A. acidoterrestris* spores by thermal processing in orange juice depends on the intensity of the treatment and the bacterial strain, presenting D values from 10 to 23 min and from 2.5 to 8.7 min at 90 and 95 °C, respectively. Therefore, the spores are resistant and can germinate and grow during storage at favorable conditions [3,17–20]. The resistance of *A. acidoterrestris* spores to high temperatures and acidic environments is due to a protective external protein coat composed of ω-alicyclic fatty acids in their membranes [21,22]. Since the inactivation of *A. acidoterrestris* spores cannot be achieved by traditional thermal processing without affecting the quality properties of the juice [23], alternative methods should be employed in order to prolong the shelf life of fruit juices and also fulfill consumer demands for minimally processed, healthy and nutritious food commodities. Alternative methods that have been used successfully to inactivate the spores of *A. acidoterrestris* include chemical treatments (e.g., ozone, chlorine dioxide, organic acids, potassium sorbate, sodium benzoate, poly dimethyl ammonium chloride); natural antimicrobial compounds of microbial origin (e.g., nisin, enterocin AS-48, Bificin C6165); natural antimicrobials of animal and plant origin (e.g., lysozyme, chitosan, essential oils); and non-thermal methods such as high hydrostatic pressure (HHP) combined with mild heat treatment, ultra-high pressure homogenization, supercritical carbon-dioxide-assisted HHP, UV-C light inactivation, irradiation, microwaves, ultrasonic waves and ohmic heating techniques [22,24,25].

High hydrostatic pressure (HPP) is an innovative processing method commercially used by many food industries, with a constantly increasing number of HHP-treated foods available on the market. It does not deteriorate food quality in terms of amino acids, vitamins, nutrients and functional properties; therefore, HHP-treated juices are of a superior quality compared with thermally processed ones [20,22]. During HHP treatment, the juice is exposed to high pressures (100–800 MPa) in order to inactivate spoilage or pathogenic microorganisms and therefore extend the shelf life of the product [12]. *A. acidoterrestris* spores are extremely resistant to HHP and the treatment must be combined with moderate heating of the juice in order to increase the effectiveness of the process [26–31].

Although the inactivation of *A. acdoterrestris* spores in fruit juices by temperature-assisted high pressure has been explored, there is little information in the literature concerning the survival of spores during storage after HHP treatment. Therefore, the objective of this study was to investigate the population dynamics of spores and vegetative cells of two *A. acidoterrestris* strains inoculated in orange juice during temperature-assisted high-pressure processing (600 MPa/60 °C for 5 and 10 min) followed by storage of the orange juice at isothermal conditions (4, 12 and 25 °C) for 2 months.

## 2. Materials and Methods

### 2.1. Bacterial Strains

Two strains of *A. acidoterrestris* were employed in this study, namely a wild-type strain (GenBank accession number MW142406) [9], kindly provided by the Laboratory of Food Microbiology and Hygiene of the Aristotle University of Thessaloniki, that was isolated previously from apple juice (denoted herewith as strain A) and a reference strain DSMZ 2498 (denoted herewith as strain B) obtained from DSMZ (Deutsche Sammlung von Mikroorganismen and Zellkuturen, Braunschweig, Germany) culture collection. Cultures of both strains were maintained at −80 °C in yeast extract starch glucose (YSG) broth (yeast extract, 2.0 g; glucose, 1.0 g; soluble starch, 2.0 g; 1000 mL $H_2O$) with pH adjusted to 3.7 using HCl (1N), supplemented with 20% (*v/v*) glycerol (APPLICHEM, Darmstadt,

Germany). To obtain the stock cultures both strains were pre-cultured in YSG broth for 48 h at 45 °C.

### 2.2. Spore Production

Fresh cultures were spread on acidified *Bacillus acidoterrestris* medium (BAT) agar plates (adjusted to pH 3.7 after sterilization with 1N $H_2SO_4$) (BTA20500, Biolab, Budapest, Hungary) and incubated at 45 °C for 7 days to sporulate. The sporulation of the cells was monitored by phase contrast microscopy and the spores were harvested when at least 80% of the cells had sporulated. Specifically, 2.5 mL of cold, sterile, distilled water was added to the BAT agar plates and the culture was gently removed from the surface by means of a sterile glass rod. The process was repeated twice and the suspensions from 15 plates were centrifuged at 5000 rpm for 20 min at 4 °C. The pellet was washed three times using cold sterile distilled water by centrifugation (5000 rpm for 20 min at 4 °C). Finally, the spores were re-suspended in 10 mL sterile phosphate buffer (pH 7.2) and stored at 4 °C until use.

### 2.3. Spore Enumeration

Volumes of 2 mL of HHP-treated (see Section 2.5) and non-treated samples were initially heat shocked at 80 °C for 10 min and then cooled in ice water in order to destroy the vegetative cells. The spore concentrations of the orange juice samples were determined from the appropriate decimal dilution after spread plating on acidified BAT agar plates (pH 3.7), incubated at 45 °C for 3 days. After incubation, plates containing 30 to 300 colonies were used for enumeration and the results were log transformed and expressed as log CFU/mL. In order to lower the detection limit of the enumeration method to 0 log CFU/mL, 1 mL of the sample was spread plated to three Petri dishes of BAT agar. It needs to be noted that microbiological analyses were undertaken before and after the heat shock treatment in order to determine the population of vegetative cells of the bacterium, by subtracting the counts before (spores and vegetative cells) and after (spores only) the heat shock treatment for HHP-treated and untreated (control) samples.

### 2.4. Orange Juice Samples

Commercially pasteurized orange juice (pH 3.7, 11.45 °Brix) was purchased from the local market. The juice was subjected to microbiological analysis to detect the presence of *A. acidoterrestris* and the results indicated absence of the bacterium from the commercial sample of orange juice. Plastic film pouches (45 mm wide × 95 mm long × 90 µm thickness) with $O_2$ permeability of 75 mL/m$^2$/24 h/1 atm at 23 °C and 75% relative humidity (Flexo-Pack SA., Athens, Greece) were filled with 4 mL of orange juice and 0.4 mL of spore suspension in order to obtain final spore concentration of ca.$10^7$ spores/mL. The pouches were heat-sealed with the use of a HenkoVac 1700 machine (Howden Food Equipment B.V., Hertogenbosch, The Netherlands) taking care to expel most of the air.

### 2.5. HPP Thermal Processing of Orange Juice Samples

The pouches were subjected to temperature-assisted high-pressure treatment at 600 MPa/60 °C for pressurization times of 5 and 10 min, respectively. The HHP treatments were conducted with a Food Pressure Unit (FPU) 1.01 (Resato International BV, Roden, The Netherlands). The system comprised a pressure intensifier and a 1.5 L vessel (7 cm diameter and 40 cm length), operating at a maximum pressure of 1000 MPa and temperature up to 90 °C, with pressure adjustable in steps of 20 MPa. The pressure transmitting fluid was polyglycol ISO viscosity class VG 15 (Resato International BV, Roden, The Netherlands). Further details of the HHP equipment can be found elsewhere [32]. The come-up rate was approximately 100 MPa per 7 s and the pressure release time was less than 3 s. Pressure and temperature were constantly monitored and recorded during the process with the use of pressure transducers and temperature transmitters. The pressure come-up and release times were not included in the reported treatment times. After HHP treatment, the pouches were stored in isothermal conditions (4, 12 and 25 °C) for two months and the population of

*A. acidoterrestris* spores was determined every week. Overall, the experiment was repeated twice with duplicate pouches analyzed for each combination of HHP treatment, bacterial strain, storage time and storage temperature.

## 3. Results and Discussion

Temperature-assisted high-pressure processing can achieve inactivation of *A. acidoterrestris* spores in fruit juices and, as temperature increases, the rate of microbial inactivation also increases [26,33,34]. Specifically, studies with HHP treatments at 600 MPa report microbial reduction in spore population of the microorganism depending on temperature and pressurization time. The effectiveness of HHP treatment at 600 MPa combined with a temperature of 60 °C for 5 and 10 min on the survival rate of *A. acidoterrestris* spores is illustrated in Figures 1 and 2, respectively. Results indicated an initial decrease of ca. 3 log cycles instantly after treatment of the wild-type strain A regardless of pressurization time, whereas the respective reduction for the reference strain B was ca. 3 and 3.5 log cycles for 5 and 10 min pressurization times, respectively (Figures S1 and S2). These results are in agreement with Hartyani et al. (2013) [12] who inoculated *A. acidoterrestris* spores in orange juice and applied high-pressure treatment at 600 MPa/60 °C for 10 min and reported a reduction of 3 log cycles in the spore population immediately after pressurization. Moreover, Vercamenn et al. (2012) [34] applied high-pressure treatment at 600 MPa/60 °C for 10 min using tomato sauce (pH 4.2) and reported a reduction of 3.5 log cycles in *A. acidoterrestris* spores. Moreover, the spore population of the wild strain A presented a change of less than 0.5 log cycles throughout storage at 4 °C for the orange juice treated at 600 MPa/60 °C for 5 min (Figure 1), while for the respective treatment for 10 min, the population showed a gradual decrease of ca. 1 log cycle up to 21 days of storage ($p < 0.05$) followed by fluctuations of ca. 0.5 log cycles. For storage at 12 °C, the spore population after the 5 min HHP treatment showed fluctuations that ranged between 0.5–1 log cycles, with the exception of 35 days where a 1.5 log cycle reduction was noticeable. For the 10 min HHP treatment, fluctuations ranged between 0.7 and 1.3 log cycles within the first 35 days and subsequently increased gradually throughout storage, exceeding the initial population by ca. 0.5 log cycles. In the case of orange juice samples stored at 25 °C, both HHP treatments (5 and 10 min) resulted in a 1 log cycle increase in spores at the end of storage. The spore population of the reference strain B (Figure 2), when stored at 4 °C, also showed fluctuations and a decrease of 1.5 log cycles below the initial counts after 60 days, regardless of treatment time. Storage at 12 °C presented different trends for the two treatments. Specifically, for the 5 min treatment a decrease of 1.5 log cycles was observed up to day 35, followed by an increase without reaching the initial counts. On the contrary, for the 10 min treatment, the population oscillated between 0.6 and 0.8 log cycles without presenting a particular trend. At 25 °C, both treatments presented ca. 1 log cycle reduction in spore counts after 1 week and subsequent fluctuations for the remaining time without reaching the initial population at the end of storage. Overall, no remarkable changes in the spore population of the wild strain A were observed after 60 days of storage of the orange juice at 4 and 12 °C, whereas an increase of 1 log cycle in *A. acidoterrestris* spores was observed after 60 days storage at 25 °C, for both HHP treatments. For the reference strain B, results also did not reveal remarkable differences throughout storage at all temperatures assayed after HHP treatment for 5 min, whereas for 10 min processing, spore population increased by ca. 1 log cycle after 60 days of storage at 12 °C, but no differences were established throughout storage at 4 and 25 °C. Hartyani et al. (2013) [12] inoculated *A. acidoterrestris* spores of the same reference strain (DSMZ 2498) in orange juice and applied HHP at 600 MPa/60 °C for 10 min followed by juice storage at 4 °C for 28 days and reported ca. 1 log cycle reduction in spore population after 28 days of storage, which is in good agreement with the results obtained in this work for the same strain (Figure 2). In order to determine the presence of vegetative cells after the HHP treatment and elucidate their behavior throughout storage at all temperatures (4, 12 and 25 °C) for both *A. acidoterrestris* strains, counts were considered before and after the heat shock treatment.

The difference represents the population of vegetative cells as shown in Figures 3 and 4 for strain A and B, respectively. The population of vegetative cells after the HHP treatment for both strains was 1.5 log CFU/mL and ca. 1.0 log CFU/mL for 5 and 10 min pressurization times, respectively. Vercamenn et al. (2012) [34] reported a 0.5 log CFU/mL population of vegetative cells when applying HHP and heat treatment under the same conditions for a different strain of *A. acidoterrestris* in tomato sauce at pH 4.2, while Hartyani et al. (2013) [12] reported a 1.0 log CFU/mL population of vegetative cells for the same treatment with the same reference strain (DSMZ 2498) in orange juice. In addition, Riberio and Cristianini (2020) [35] reported a 1.24 log CFU/mL population of vegetative cells after HHP treatment at 600 MPa/70 °C/5 min in phosphate buffer medium (pH 7.2).

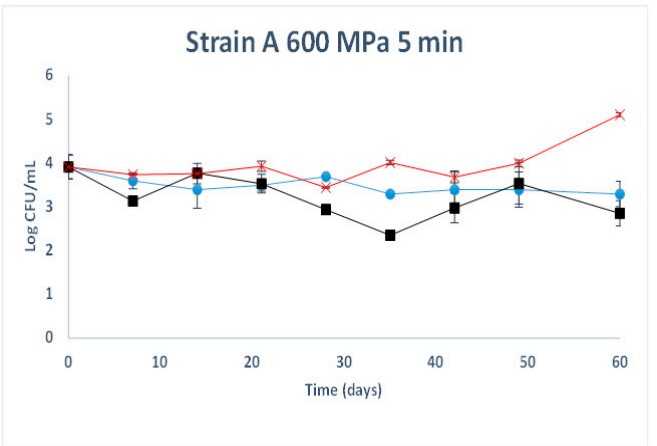
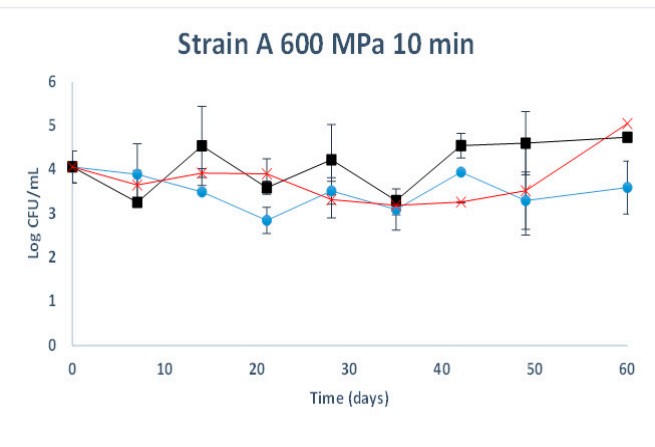

**Figure 1.** Survival curves of *A. acidoterrestris* (Strain A) spores treated at 600 MPa/60 °C for 5 and 10 min and stored at 4 °C (●), 12 °C (■) and 25 °C (×). Data points are mean values ± standard deviation of duplicate samples from two independent experiments (*n* = 4).

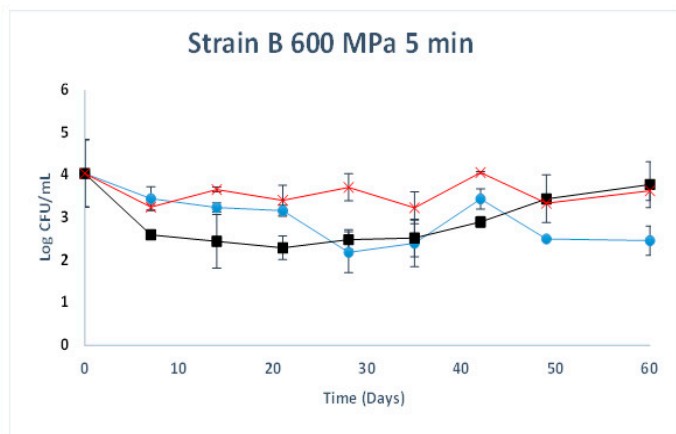
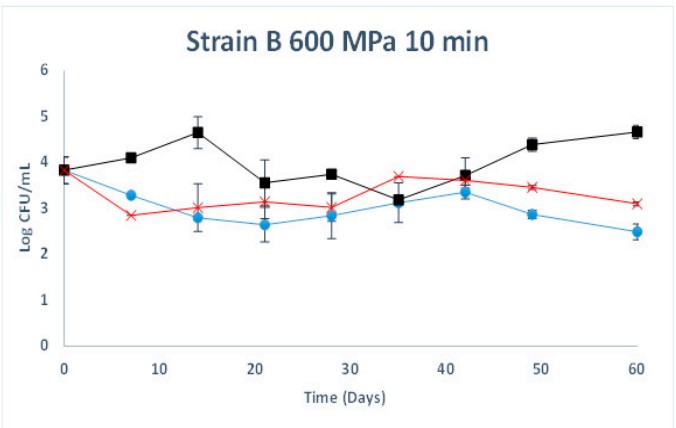

**Figure 2.** Survival curves of *A. acidoterrestris* (Strain B) spores treated at 600 MPa/60 °C for 5 and 10 min and stored at 4 °C (●), 12 °C (■) and 25 °C (×). Data points are mean values ± standard deviation of duplicate samples from two independent experiments (*n* = 4).

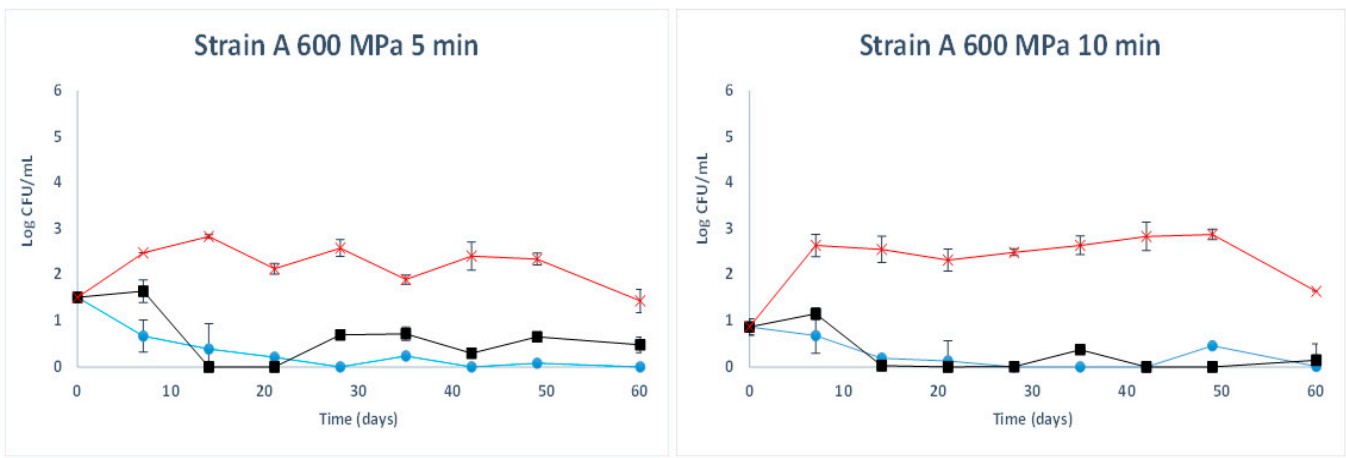

**Figure 3.** Survival curves of *A. acidoterrestris* (Strain A) vegetative cells treated at 600 MPa/60 °C for 5 and 10 min and stored at 4 °C (●), 12 °C (■) and 25 °C (×). Data points are mean values ± standard deviation of duplicate samples from two independent experiments (*n* = 4).

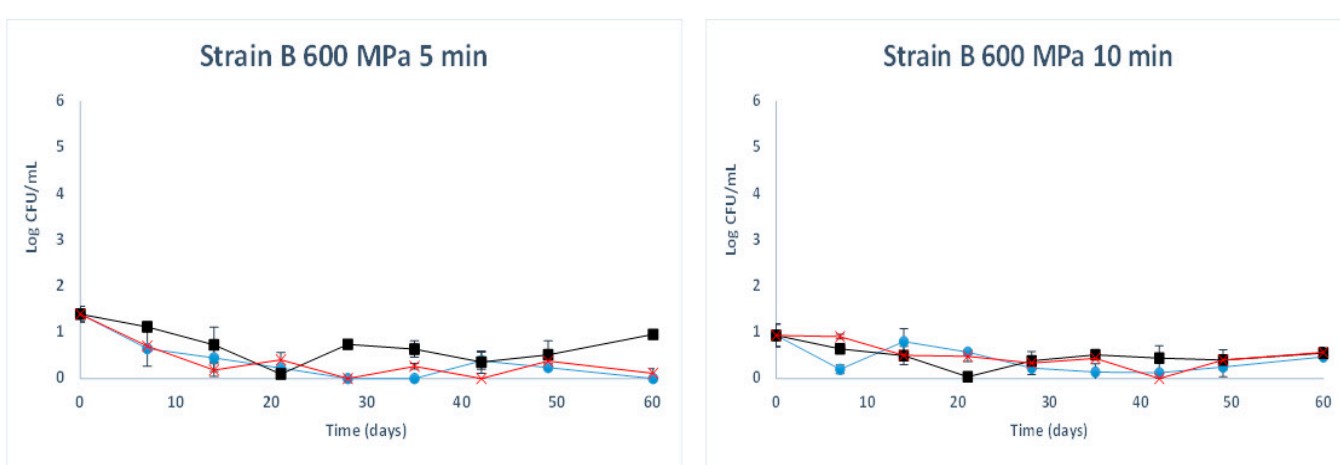

**Figure 4.** Survival curves of *A. acidoterrestris* (Strain B) vegetative cells treated at 600 MPa/60 °C for 5 and 10 min and stored at 4 °C (●), 12 °C (■) and 25 °C (×). Data points are mean values ± standard deviation of duplicate samples from two independent experiments (*n* = 4).

However, at 12 °C, the population of vegetative cells was reduced after day 7, whereas at 4 °C their population reduced throughout storage and was found near the detection limit after day 21 for both HHP treatments of 5 and 10 min. Consequently, vegetative cells can be eliminated with HHP treatment during storage at low temperatures, which is in agreement with Hartyani et al. (2013) [12], who reported that the population of vegetative cells of *A. acidoterrestris* treated with HHP at three different pressure levels (200, 400 and 600 MPa) at 60 °C for 10 min in orange and apple juice was decreased to the detection limit of the plating method throughout storage at 4 °C for 28 days. As shown in Figure 3, the counts of vegetative cells of strain A were increased and maintained above the initial population throughout storage at 25 °C for 60 days. On the contrary, the vegetative cells of strain B did not follow the same trend, since the population decreased even at 25 °C. This difference between the wild-type strain A and the reference strain B could be attributed to the fact that the effectiveness of the HHP treatment against *A. acidoterrestris* is strain-dependent [36], presumably due to the different distribution of fatty acids in the membranes of the bacteria [37], although other factors could be cell age, cell population and nutrient availability [38,39].

## 4. Conclusions

In conclusion, the results from this study indicated that HHP could induce the inactivation of *A. acidoterrestris* spores in orange juice when combined with mild heat treatment and ensure the inhibition of surviving spore germination during the shelf life of the final product at refrigerated temperatures. However, during storage at different isothermal conditions for two months, the changes in spore population were strain-dependent and did not present a consistent pattern. It was also demonstrated that the remaining vegetative cells could be eliminated throughout storage at low temperatures, although strain variability should be taken into consideration. Therefore, heat-assisted HHP treatment could offer promising perspectives for reducing spoilage and extending the shelf life of orange juice.

**Supplementary Materials:** The following supporting information can be downloaded at: https://www.mdpi.com/article/10.3390/fermentation8070308/s1, Figure S1: Survival curves of non-HHP-treated (control) *A. acidoterrestris* spores of Strain A and Strain B stored at 4 °C (•), 12 °C (■) and 25 °C (×). Data points are mean values ± standard deviation of duplicate samples from two independent experiments ($n$ = 4), Figure S2: Survival curves of non-HHP-treated (control) *A. acidoterrestris* vegetative cells of Strain A and Strain B stored at 4 °C (•), 12 °C (■) and 25 °C (×). Data points are mean values ± standard deviation of duplicate samples from two independent experiments ($n$ = 4).

**Author Contributions:** Conceptualization, P.S., A.A.A., G.-J.E.N. and C.C.T.; methodology, P.S. and A.A.A.; formal analysis, P.S. and E.Z.P.; investigation, P.S.; resources, C.C.T. and G.-J.E.N.; writing—original draft preparation, P.S.; writing—review and editing, P.S., A.A.A., C.C.T. and E.Z.P.; supervision, C.C.T., G.-J.E.N. and E.Z.P. All authors have read and agreed to the published version of the manuscript.

**Funding:** This research received no external funding.

**Institutional Review Board Statement:** Not applicable.

**Informed Consent Statement:** Not applicable.

**Data Availability Statement:** The data presented in this study are available on request from the corresponding author.

**Acknowledgments:** Patra Sourri would like to thank the Hellenic Agricultural Organization DIMITRA for supporting this work as part of her Ph.D. thesis.

**Conflicts of Interest:** The authors declare no conflict of interest.

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
