# Peer review of "The Effect of Temperature-Assisted High Hydrostatic Pressure on the Survival of Alicyclobacillus acidoterrestris Inoculated in Orange Juice throughout Storage at Different Isothermal Conditions"

_fermentation, doi:10.3390/fermentation8070308_

Round 1

Reviewer 1 Report

In the last decade, the high hydrostatic pressure approach has been extensively studied to inactivate bacteria and spores in food products and commercial juices, namely Alicyclobacillus acidoterrestris. Therefore, the process proposed of combined high hydrostatic pressure and heat treatment to inactivate spores in juices is not innovative enough, the experimental data are insufficient and does not provide relevant data. So, it is recommended not to be accepted.

Author Response

Comment: In the last decade, the high hydrostatic pressure approach has been extensively studied to inactivate bacteria and spores in food products and commercial juices, namely Alicyclobacillus acidoterrestris. Therefore, the process proposed of combined high hydrostatic pressure and heat treatment to inactivate spores in juices is not innovative enough, the experimental data are insufficient and does not provide relevant data. So, it is recommended not to be accepted.

Response: We are aware that High Hydrostatic Pressure has been studied extensively with special focus on spore inactivation immediately after HHP treatment of fruit juices. To the best of our knowledge there is little information concerning the dynamics of spores and vegetative cells of HHP treated Alicyclobacillus acidoterrestris after treatment and during storage. We believe that the innovation in this work lies in the investigation of the survival of spores and vegetative cells during storage of orange juice at different temperature regimes. This aspect was not clear in the initial version of our manuscript as the aim of our research was accidentally excluded. We have now revised this part by including the purpose of this work in the last paragraph of the introduction.

Reviewer 2 Report

The manuscript descries the investigation of the population dynamics of the spores and vegetative cells of A. acidoterrestris in orange juice which has been treated with temperature assisted HHP and stored under different condition. The manuscript is well written and the scientific objectives are addressed adequately by the experimental design. I therefore recommend its publication. 

Author Response

Comment: The manuscript descries the investigation of the population dynamics of the spores and vegetative cells of A. acidoterrestris in orange juice which has been treated with temperature assisted HHP and stored under different condition. The manuscript is well written and the scientific objectives are addressed adequately by the experimental design. I therefore recommend its publication.

Response: We would like to thank the reviewer for his/her positive approach to our work.

Reviewer 3 Report

This work is related to inactivate A. acidoterrestris in orange juice using heat assisted HHP treatment. Authors should answer the points below carefully to put article into publishable form.

1. What are the numbers of spores and vegetable cells in the initial inoculum? According to the Figures in the Supplementary Materials, there are no vegetable cells in the initial inoculum?

2. Please explain the setting method of control samples in the text.

3. Please add the information of detection limit of the plate count method.

4. The author concluded that heat assisted HHP treatment could ensure the inhibition of the surviving spores’ germination throughout storage at low temperatures. But in the control sample, the spore didn’t germinate as well during the shelf-life of the final product at refrigerated temperatures. Therefore, I think the statement of the conclusion is not rigorous.

Author Response

Comment 1: What are the numbers of spores and vegetable cells in the initial inoculum? According to the Figures in the Supplementary Materials, there are no vegetable cells in the initial inoculum?

Response: As shown in the supplementary figure the initial population of the spores in the control samples was ca. 7 log CFU/mL and there were no vegetative cells for both A. acidoterrestris strains. This verifies our initial intention to produce an inoculum containing spores.

Comment 2: Please explain the setting method of control samples in the text.

Response: A comment on this issue has been added in the revised manuscript. Please see page 3, line 132.

Comment 3: Please add the information of detection limit of the plate count method.

Response: The information has been added in lines127-128 of page 3.

Comment 4: The author concluded that heat assisted HHP treatment could ensure the inhibition of the surviving spores’ germination throughout storage at low temperatures. But in the control sample, the spore didn’t germinate as well during the shelf-life of the final product at refrigerated temperatures. Therefore, I think the statement of the conclusion is not rigorous.

Response: HHP induces the germination of spores which are subsequently inactivated by temperature. However, in the control samples the conditions were not suitable (no pressurization) to support the germination of the spores at refrigerated temperatures.

Reviewer 4 Report

The manuscript entitled “The effect of temperature assisted high hydrostatic pressure on the survival of Alicyclobacillus acidoterrestris inoculated in orange juice throughout storage at different isothermal conditions” aims, on the one hand, to evaluate the lethality of two strains (wild and collection strains) inoculated in the juice of orange and treated at 600 MPa for 5 and 10 min. Furthermore, it was evaluated their growth during storage at 4, 12 and 25 °C, obtaining a juice with a certain equivalence to a pasteurization treatment. In the lethality study, the results are similar to those published by these authors (Sourri et al., 2020). In this study, differences in the effect of the  HPH treatment were already evident depending on the type of strain and the HPH treatment performed (pressure, temperature, and tine). A new contribution of this manuscript would be the study during its conservation of both the sporulated and vegetative microbiota. As expected, and as it happens in pasteurized products, it is demonstrated that the microorganism has a low capacity for growth, with a relatively relevant increase in the study at 25 °C.

Regarding the experimental design, it could be considered that conducting the study twice would not be entirely sufficient to obtain conclusions. The authors present an N=4, but only two experiments with a duplication of the analysis were performed. This number of samples would not be enough to apply an ANOVA test or to study the effect of the survival factors and their modelization. Then, Figure 5 would not contribute practically anything to explain these factors. In this sense, Tables 1 and 2 would not provide relevant information, and the prediction equation would not be very constable.

The authors explain in detail the growth of the spores. However, the results could be simplified as no growth occurs regardless of pressure, temperature, and strain when stored at 4 and 12 °C, except for a slight increase in the development of Alicyclobacillus in the treatments of 600 MPa 5 min and 10 min in the wild strain. Of much more interest is the survival of vegetative forms, in which HHP and low temperatures seem to have a damaging effect, especially in the wild strain. It would be very interesting to know the reason for these differences between strains.

An issue also related to the design is why pressure and times were selected, considering the previous study performed by these authors?

In short, although the manuscript provides relevant findings, it could be summarized many more.

 (1)    Sourri, P., Argyri, A. A., Panagou, E. Z., Nychas, G. J. E., & Tassou, C. C. (2020). Alicyclobacillus acidoterrestris strain variability in the inactivation kinetics of spores in orange juice by temperature-assisted high hydrostatic pressure. Applied Sciences (Switzerland), 10(21), 1–11. https://doi.org/10.3390/APP10217542

Author Response

Comment 1: The manuscript entitled “The effect of temperature assisted high hydrostatic pressure on the survival of Alicyclobacillus acidoterrestris inoculated in orange juice throughout storage at different isothermal conditions” aims, on the one hand, to evaluate the lethality of two strains (wild and collection strains) inoculated in the juice of orange and treated at 600 MPa for 5 and 10 min. Furthermore, it was evaluated their growth during storage at 4, 12 and 25 °C, obtaining a juice with a certain equivalence to a pasteurization treatment. In the lethality study, the results are similar to those published by these authors (Sourri et al., 2020). In this study, differences in the effect of the HPH treatment were already evident depending on the type of strain and the HPH treatment performed (pressure, temperature, and tine). A new contribution of this manuscript would be the study during its conservation of both the sporulated and vegetative microbiota. As expected, and as it happens in pasteurized products, it is demonstrated that the microorganism has a low capacity for growth, with a relatively relevant increase in the study at 25 °C.

Regarding the experimental design, it could be considered that conducting the study twice would not be entirely sufficient to obtain conclusions. The authors present an N=4, but only two experiments with a duplication of the analysis were performed. This number of samples would not be enough to apply an ANOVA test or to study the effect of the survival factors and their modelization. Then, Figure 5 would not contribute practically anything to explain these factors. In this sense, Tables 1 and 2 would not provide relevant information, and the prediction equation would not be very constable.

Response: We would like to thank the reviewer for this comment. We had extensive discussions with the Statistical department of the Agricultural University of Athens on this issue and we were informed that there are no precise rules on this aspect. The statisticians supported that what is most important in the ANOVA analysis is the number of degrees of freedom in the Residual, which in our case is 309. To have a robust ANOVA analysis we need a high value for F (in our case it was 284.68). However, we would agree with the reviewer that Figure 5 does not contribute practically to our work and thus we decided to exclude it from the revised manuscript. 

Comment 2: The authors explain in detail the growth of the spores. However, the results could be simplified as no growth occurs regardless of pressure, temperature, and strain when stored at 4 and 12 °C, except for a slight increase in the development of Alicyclobacillus in the treatments of 600 MPa 5 min and 10 min in the wild strain. Of much more interest is the survival of vegetative forms, in which HHP and low temperatures seem to have a damaging effect, especially in the wild strain. It would be very interesting to know the reason for these differences between strains.

Response: A general description of the results has been added after the detailed explanation on page 6,  lines 218-225 in order to summarize the results and make them clearer. Regarding the differences between the two strains, a comment has been added in the revised manuscript (page 7, lines 265-266), although this aspect was beyond the scope of this work,  

Comment 3: An issue also related to the design is why pressure and times were selected, considering the previous study performed by these authors?

Response: Although the selected times and pressures did not show significant differences in the previous study, we aimed to investigate the presence of any differences during storage of orange juice at different temperatures. This particular pressurization conditions are also more relevant to juice processing at industrial level.

Round 2

Reviewer 1 Report

The manuscript quality has improved after review process and now the purpose of this work is more clearly.

A minor aspect should be modified: 

Line 57: please correct A. acdoterrestris for A. acidoterrestris

Author Response

Comment: The manuscript quality has improved after review process and now the purpose of this work is more clearly. A minor aspect should be modified: Line 57: please correct A. acdoterrestris for A. acidoterrestris.

Response: Corrected.

Reviewer 4 Report

The manuscript has generally improved substantially after adequately responding to suggestions arising from the review process.

However, as previously mentioned, data at 4 and 12 °C were predictable, considering the thermophilic character of Alicyclobacillus. What is interesting in this article is the germination effect of the HHP treatment and the evaluation of whether storage at 25 ºC favours its growth. Moreover,  the experimental design carried out and considering that the storage temperature would not be a factor to be considered, the proposed general linear model with interactions does not provide relevant information to the study.

Author Response

Comment: The manuscript has generally improved substantially after adequately responding to suggestions arising from the review process. However, as previously mentioned, data at 4 and 12 °C were predictable, considering the thermophilic character of Alicyclobacillus. What is interesting in this article is the germination effect of the HHP treatment and the evaluation of whether storage at 25 ºC favours its growth. Moreover, the experimental design carried out and considering that the storage temperature would not be a factor to be considered, the proposed general linear model with interactions does not provide relevant information to the study.

Response: Based on the comment on the relevance of the general linear model, we have decided to exclude this part from the revised manuscript.